# How Do Environmental Technology Standards Affect the Green Transformation? New Evidence from China

**DOI:** 10.3390/ijerph19105883

**Published:** 2022-05-12

**Authors:** Xiufeng Zhang, Yan Li, Ke Shi, Yanchao Feng

**Affiliations:** 1Business School, Henan Normal University, Xinxiang 453007, China; zhangxiufeng2003@163.com; 2Business School, Zhengzhou University, Zhengzhou 450001, China; shike_zzu@stu.zzu.edu.cn (K.S.); fengyanchao@zzu.edu.cn (Y.F.); 3Post-Doctoral Mobile Research Station of School of Politics and Public Management, Zhengzhou University, Zhengzhou 450001, China

**Keywords:** environmental technology standard, green transformation, technological modification, porter hypothesis, manufacturing industry

## Abstract

During the period of high-quality development in China, environmental regulations have been regarded as powerful exogenous forces, especially for accelerating the green transformation of the manufacturing industry. Treating the first implementation of cleaner production industry standard policies in 2003 as a quasi-natural experiment, and employing the difference-in-differences (*DID*) model, this paper discusses the impacts of environmental technology standards on the green transformation of the manufacturing industry in China, with systematic consideration of three technological modification mechanisms, including terminal governance, capital renewal, and resource structure adjustments. The results show that: (1) environmental technology standards can have “win–win” effects, i.e., environmental technology standards can simultaneously improve the environmental performance and the economic performance; (2) the dynamic effects of environmental technology standards are proven in the parallel trend test; (3) environmental technology standards can not only promote the green transformation of the manufacturing industry directly, but can also mediate the effects of terminal governance, capital renewal, and resource structure adjustments. In summary, this paper makes four suggestions for government actions, namely re-examining command-and-control environmental regulations, paying attention to environmental regulation tools, changing the mechanism from technological modification to technological innovation, and updating environmental technology standards regularly.

## 1. Introduction

Over the past 40 years, China has quickly completed the century-old industrialization path of developed countries based on its “late-comer advantage”, and its role as “workshop to the world” has helped China achieve dual-growth miracles in terms of speed and time, resulting in the rapid accumulation of material wealth [1]. However, we all know that the past economic achievements in China were made with sacrifices in terms of resources and environmental protection [1,2,3,4]. Against this background, in order to break the “environment pollution–economic development” cycle and reduce the risk of falling into the “middle-income trap”, China’s economic goal has changed from “high-speed” to “high-quality”, which indicates that the driving force of traditional extensive development focusing on resource inputs and energy consumption is weakening [5], and a new intensive development mode focusing on environmental protection and resource conservation is becoming the norm.

In response to this new economic development mode, enterprises are required to concentrate on green transformation, especially manufacturing enterprises. Green transformation, in terms of content, refers to the reengineering of the whole production process of enterprises based on the standards related to resource conservation and environmental protection; that is to say, the key difference between green growth and traditional growth lies in whether environmental performance is considered. As a result of this consideration, the economic performance evaluation criterion has gradually changed from GDP to green GDP [6]. In addition, scholars have proposed the environmental total factor productivity (ETFP) by incorporating environmental factors into the measurement of total factor productivity (TFP) [7,8]. Generally speaking, green transformation, as a change, results in additional costs for enterprises, which directly hinders their voluntary reform. Therefore, government intervention, such as through environmental regulations, has become the inevitable choice to guide enterprises to get rid of the existing development model and strive to achieve green growth. Meanwhile, the use of environmental regulations has been proven to be the most direct and effective means for government to ensure the green transformation of enterprises [9,10]. Against this background, there are some valuable questions that must be answered: Do environmental regulations affect the green transformation of manufacturing enterprises? How do environmental regulations affect the green transformation of manufacturing enterprises? Trying to answer the above questions is the first motivation of this thesis.

Moreover, it is obvious that prior studies have not reached consistent conclusions, which leaves a lot of potential space for future research. Firstly, since the previous studies have mainly focused on developed countries, studies in the context of China are urgently called for. In fact, compared with developed countries with a more systematic environmental regulation system, the relevant systems in China still need to be perfected; accordingly, the experience generated by developed countries may not suitable in China. Secondly, although the existing studies have deeply studied the impacts of environmental regulation by divided it into three broad categories—command-and-control, market-based, and voluntary [2,11]—there is still a basic truth that not all kinds of environmental regulations and environmental regulation tools have received equal attention [11], especially in the context of China, where the environmental regulation system has not been completed. Specifically, scholars used to advocate the advantages of market-based environmental regulations and ignore command-and-control environmental regulations in the past. Even in the limited relevant research, scholars have mainly discussed the command-and-control environmental regulation approach from the perspective of the total amount of control, rather than the perspective of the technology standards. Therefore, to obtain a more objective understanding of environmental regulations and draw more targeted and specific conclusions, it is necessary to increase the research on environmental technology standards based on a more in-depth classification of environmental regulations in the future. Thirdly, compared with existing studies that often use regional- and industry-level data for empirical analyses, enterprise-level studies, which can reflect the micro-effects and mechanisms of environmental regulations, deserve deeper exploration. In summary, taking into account the above three kinds of problems in the relevant research is another motivation of this study.

Meanwhile, according to Porter’s hypothesis, technological innovation is considered an effective way to achieve green transformation, and this should encourage more scholars to study the functions of technological innovation between environmental regulations and green transformation. The core point of Porter’s hypothesis, consistent with the ecological modernization theory, is that well-designed environmental regulations can simultaneously improve economic performance and reduce pollution emissions by overcoming the inertia of technological innovation [12]. However, a large number of examples have shown that technological innovation is not the inevitable result of all enterprises responding to environmental regulations. On the contrary, when considering various constraints, technological modifications, which to completely different to technological innovation and mainly relate to the technological results when an enterprise does not directly participate in the R&D process, is the best and first choice for enterprises [13,14,15]. For example, in view of the higher investment requirements, higher costs, and longer time periods involved in technological innovation than technological modifications, enterprises, especially those with insufficient resource endowments, tend to choose technological modifications rather than technological innovation to achieve the goal of environmental regulation at low cost and with high efficiency. Unfortunately, in the past, there have been few studies on technological modifications; that is, scholars have ignored the technological modification mechanism of environmental regulation, which is worthy of further study and forms the third motivation of this paper.

To sum up, on the basis of breaking through the existing research limitations, this paper mainly makes two contributions. First, this paper creatively and empirically discusses the relationship between environmental technology standards and the green transformation of China’s manufacturing industry from the perspective of environmental regulation tools, based on an integrated methodological framework including dynamic and static aspects at the enterprise level. Second, this study selects terminal governance, capital renewal, and resource structure adjustments as the main mediating variables to investigate the effects of environmental technology standards on the green transformation from the perspective of technological modifications, providing new perspective for related research.

The remainder of this paper is organized as follows. The literature review is presented in Section 2. The methodology is shown in Section 3. The empirical results and analysis are displayed in Section 4. Section 5 aims to concludes the research, address certain policy implications, and list some of the research prospects. Additionally, a flowchart diagram of the research steps of this thesis is shown in Figure 1.

## 2. Literature Review

### 2.1. The Effects of Environmental Technology Standards

To date, there have been few significant attempts to investigate the environmental technology standards, although as a typical command-and-control type of environmental regulation, the effects must undoubtedly be in line with those of the environmental regulation approach, which are categorized into two aspects from a static perspective. On the one hand, according to the “compliance cost effect”, environmental technology standards are not conducive to the green transformation of enterprises, because strict environmental regulations always impede the economic growth potential of enterprises in the short term by crowding out operation or innovation expenditures and increasing the costs of green management [10,16,17,18]; under the pressure of environmental regulation, enterprises often achieve low-level green development by sacrificing economic performance. Based on this point of view, a large number of studies have been carried out. For instance, Boyd and McClelland [19], using data from the US paper industry, empirically proved that enterprises’ expenditure on pollution abatement crowded out other effective investments, significantly weakening the performance of the industry as a whole. As the same time, Zhao [20] clearly pointed out that environmental regulation seriously hindered the technological innovation of regulated firms by crowding out the funds that were originally used for R&D. Moreover, Zhao et al. [21] claimed that with the strengthening of environmental pollution control, the corresponding cost expenditure of firms increases, which ultimately affects the competitiveness of firms. Yuan and Xiang [22] found that innovation, in the long run, is often limited by the cost of complying with environmental regulations; that is, firms, when facing various constraints on economic growth, should cut R&D costs in response to the increasingly strict regulations. On the other hand, environmental technology standards, under the theoretical framework of the “innovation compensation effect”, are expected to promote the green transformation of enterprises. In essence, the innovation compensation effect, which is directly derived from Porter’s hypothesis, mainly states that well-designed environmental regulations can be beneficial to the innovation of enterprises, meaning a “win–win” effect can be achieved in the long run. As such, considerable research has supported this view. Specifically, Desrochers and Haight [23] claimed that environmental regulation is helpful for enterprises to find the best technological transformation approach, and this ultimately improves efficiency in technical innovation and the level of TP. Li et al. [24] revealed that environmental regulation, by creating absolute advantages, including “innovation advantages” and “first-mover advantages”, may improve a firm’s financial performance. Liu et al. [25] stated that by facilitating technological innovation, environmental information disclosure can increase the economic performance of regulated enterprises.

However, despite continuous attention being paid to the issue of environmental regulation, the practical results cannot be properly explained by the theoretical results, which has forced scholars to reflect on the limitations of the above two kinds of effects, which to a certain extent do provide the basis and inspiration for the non-linear effects of environmental technology standards. In short, the “compliance cost effect” does not take into account the possibility of changes in corporate behaviors brought about by gradually increased awareness of the importance of environmental protection [26], while the “innovation compensation effect” causes conflict between the assumption of enterprise inefficiency and the assumption of an enterprise’s pursuit of profit maximization. Against this background, there is a preliminary consensus that the “compliance cost effect” and “innovation compensation effect” often coexist in a state of asynchrony [5,10], which has directly led to the popularity of the dynamic analysis approach. In detail, in the beginning phase of the implementation of environmental regulations, enterprises have limited willingness to carry out innovation because of their awareness of environmental regulations and the consequences for failing to adhere to them. Eventually, the “compliance cost effect” dominates at this stage and its negative effect is emphasized. Conversely, after the implementation of environmental regulations for a period of time, the strengthened awareness and consequences force enterprises to seek change through innovation. Finally, the “innovation compensation effect” becomes dominant and its positive role is highlighted. Enlightened by the above views, scholars have conducted extensive relevant research. Among them, Zhang and Wei [27] concluded that there is a non-linear relationship between environmental regulation and carbon emissions, which is described by an inverted U-shape. Similarly, Wang and Shen [28] empirically verified the existence of an inverted U-shaped non-linear relationship between environmental regulation and green productivity. Nevertheless, Guo et al. [18] posited that there is an inflection point in the relationship between environmental regulation and GTI, which finally forms a U-shaped curve.

In addition, both static and dynamic analysis results are significantly affected by regulatory variables including the environmental regulation, pollutant, enterprise, industry, region, and other aspects, which increases the complexity of the research results. Correspondingly, plenty of studies have emerged. For instance, Lin et al. [29] suggested that when considering the differences in governance models and innovation characteristics, enterprise ownership is one of the factors that affect the relationship between environmental regulation and green technological innovation. In detail, state-owned firms, in contrast, are less likely to avoid government intervention based on their unique organizational goals. Lee et al. [30] stated that the level of environmental regulation is one of the important factors affecting GTI and subsequent technological changes. Furthermore, the higher the level of environmental regulation, the greater the contribution to GTI and the impacts on the subsequent technological changes. Greenstone et al. [31] confirmed that the impacts of environmental regulations are determined by the kinds of regulated pollutants, drawing the conclusion that ozone and particulate emission regulations have negative effects, sulfur dioxide emission regulations have no effect, and carbon monoxide regulations have positive effects. Zhao et al. [21] systematically discussed the effects of three kinds of environmental regulations, and the results illustrated that different environmental regulations show different effects on the same enterprise objective. Feng and Chen [12] pointed out that from the perspective of regional differences, the effects of environmental regulation on industrial green development at the regional level are unevenly distributed.

As mentioned above, the impacts of environmental regulation are complex enough for scholars to truly achieve the goal of guiding practice with theory, which requires a nearly perfect study design. Against this background, in order to accurately identify the effects of environmental technology standards on green transformation, this thesis is required to consider not only static and dynamic analysis, but also China’s local context, the classification of environmental regulation tools, and the types of industries in which the enterprises are located.

### 2.2. The Mechanism of Environmental Technology Standards

Most existing studies, based on Porter’s hypothesis, have confirmed that technological innovation is the main mechanism of environmental regulation, which not only reflects the reality to some extent, but also shows a good expectation of people’s behavior after they fully understand the long-term advantages of technological innovation. However, it should be noted that the relevant research is not completely consistent with Porter’s hypothesis (environmental regulation must be conducive to the development of enterprises through technological innovation); on the contrary, scores of documents have shown that even if the intermediary mechanism of technological innovation is determined, the effect of the mechanism on enterprises is not certain [22,32]. Correspondingly, Lanoie et al. [33] found that although strict environmental regulation triggered technological innovation in enterprises, the impacts on the economic performance of the enterprise were still negative, because the innovation compensation effect was not accounted for or even exceeded the compliance cost effect. Rubashkina et al. [32] demonstrated that it is precisely because of the innovation effect—which can make the production of products and processes more efficient—that environmental regulation becomes a “win–win” strategy. Franco and Marin [34] pointed out that environmental protection taxes have significantly promoted technological innovation as measured by patents at the industry level, thereby increasing industry productivity. Guo et al. [35] stated that only when driven by technological innovation can environmental regulation be conducive to regional green growth. Ren et al. [36] concluded that technological innovation is an important mechanism to explain the “win–win” effect of China’s pilot areas of emissions trading. Deng et al. [37] verified the positive effects of China’s “energy conservation and low carbon” policy by the conclusion that the innovation compensation effect is significantly greater than the compliance cost effect.

However, the view that technological innovation is the only mechanism of environmental regulation has been rejected, with the support of practices, especially those from developing economies, also being important. Furthermore, technological modification has been considered as another popular mechanism [15]. Specifically, Ashford [38] and Renning et al. [39] elaborated on the effectiveness and superiority of integrated technologies in corporate environmental compliance, and Ashford [38] pointed out that integrated solutions often concentrate on pollution prevention based on their “forward-looking, anticipate, and prevent philosophy”; additionally, due to the possibility of saving costs by reducing the use of raw materials and energy, the costs of using integrated technology are lower than those of end-of-pipe solutions. Liu [40] clearly showed that Chinese firms in the past always choose end-of-pipe treatments to meet the general principles of green development, and one of the direct consequences of this behavior is that more decontamination equipment is introduced into firm’s production chain. Wang and Shen [28] claimed that enterprises are not always able to deal with environmental regulation through technological innovation, but choose to do so dynamically according to the intensity of the environmental regulation. Generally speaking, the lower the degree of environmental regulation, the stronger the willingness of people to accept punishment or choose a method of technological modification with lower costs. Moreover, they also found that the effects of environmental regulation vary with the type of enterprise, which compared with clean enterprises, the environmental regulation approach has a better effect on polluting enterprises. Xie et al. [11] suggested that discovering new end-of-pipe techniques, increasing resource efficiency, and switching to clean energy are effective pollution reduction measures taken by enterprises to improve efficiency when they facing strict environmental standards. Yuan and Xiang [22] proved that non-invention patents can help the manufacturing industry establish a comprehensive and coordinated development pattern. On this basis, government should guide the manufacturing industry to introduce energy-saving and emission reduction facilities for green process reform. Fan et al. [13] empirically found that polluting enterprises mainly meet more stringent environmental regulation requirements by increasing investment in recycling and emission reduction equipment when experiencing China’s emission reduction plan. Shi and Li [14] believed that the method of upgrading production equipment and then absorbing advanced production technology has always been an important path in the technological progress and productivity improvement of China’s manufacturing industry, which may be an effective choice to achieve green transformation using environmental regulations. Yi et al. [10] concluded that because of the “compliance cost”, not only will the innovation activities of enterprises be crowded out, but also the production strategy of enterprises will also be correspondingly changed by implementing cutbacks, layoffs, and even closures.

To sum up, although the mechanism of environmental regulation has expanded from the path of technological innovation to the path of technological modification, this new path has only been recognized at the theoretical level and lacks corresponding empirical testing. Accordingly, this paper tries to makes up for this vacancy by selecting terminal governance, capital renewal, and resource structure adjustment as the three performance measures of technological modification and carrying out systematic empirical research.

## 3. Methodology

### 3.1. Selection of Variables

#### 3.1.1. Dependent Variables

Obviously, this paper aims to explore the effects of environmental technology standards on green transformation in Chinese manufacturing enterprises and their mechanisms simultaneously, which means that this paper has two dependent variables: one is the green transformation effect and the other one is the technological modification mechanism.

Firstly, when studying the effects of environmental technology standards on green transformation, the dependent variable is green transformation. In line with the previous studies, this paper also regards green transformation as a realization of sustainable development in terms of results and adherence to the principles of resource conservation and pollution reduction in the process. However, differing from the exist studies, this paper innovatively adopts a new method of measuring economic performance and environmental performance simultaneously to show the effects on the green transformation, which can greatly help people to clearly identify the pure green contributions of environmental technology standards rather than trying to build a comprehensive indicator. Additionally, total factor productivity (TFP), which is a natural logarithm, is employed as the proxy indicator of economic performance in two ways on the basis of referring to Olley and Pakes [41] and Levinsohn and Petrin [42], ultimately marked as “*TFP_OP*” and “*TFP_LP*”, respectively, while the pollution emissions intensity, which is measured via the chemical oxygen demand emissions per unit of gross output value and industrial sulfur dioxide emissions per unit of gross output value, acts as the proxy for green transformance, for which the corresponding indicators are marked as “*COD*” and “${\mathrm{SO}}_{2}$”, respectively.

Secondly, technological modification, which is defined as all changes in an enterprise except technological innovation, is the dependent variable in the study of the mechanism of environmental technology standards. Technology modification, as a broad concept, needs to be further decomposed in this paper, and in terms of the selection of concrete mechanisms, we mainly base this on the systematic characteristics of green transformation and construct an “input–production–output” mechanism system, which involves the whole production process. On this basis, in line with the existing studies, we respectively use energy structure, capital renewal, and terminal governance as the proxy variables corresponding to the above mechanism. Furthermore, considering the data, terminal governance is mainly reflected in the capacity and quantity of various pollutant treatment instruments, including waste gas, sulfur dioxide and industrial waste water systems, the corresponding indicators for which are marked as “*GEQ*”, “*SEQ*”, “*WEQ*”, “*GCA*”, “*SCA*”, and “*WCA*”, respectively. Capital renewal is mainly reflected by three types of indicators: fixed asset investment (defined as the logarithm of the investment scale of fixed assets), depreciation (including the depreciation amount and depreciation rate), and capital efficiency (measured by dividing the average output value by the rate of expenditure over the same period), which are marked as “*INV*”, “*DEP*”, “*DEP_R*”, and “*EFF*”, respectively. Resource structure adjustment is mainly described through the use of clean energy and resource recycling by referring to Han et al. [43]. The use of clean energy is measured by the proportion of natural gas in the total consumption of fossil energy, while the recycling of resources is measured by the proportion of renewable water consumption in the total industrial consumption. Ultimately, the corresponding indicators are marked as “*CLE*” and “*REC*” in turn.

#### 3.1.2. Key Explanatory Variables

Undoubtedly, with the use of the *DID* model, we need to introduce two dummy indicators, *post* and *treat*, to help us to construct the key explanatory variables of this study, namely environmental technology standards, which are finally reflected by the interaction term (i.e., *treat × post*). Moreover, *post* represent the time information, while *treat* represent the group information. In particular, *post* equals 1 after 2003 and 0 otherwise; *treat* equals 1 when the firm *i* is in the treatment group and 0 otherwise.

#### 3.1.3. Control Variables

Except for the key explanatory variables, this study also introduces several control variables to capture certain hidden effects, which are often not directly reflected in the research objectives. Learning from the previous studies, the control variables include the enterprise size (*SZ*, defined by the natural logarithm of the total assets), external subsidies (*ES*, defined by the dummy variable, which is assigned 1 when the company receives additional subsidies and 0 otherwise), enterprise ownership (*EO*, defined by the dummy variable, which is assigned 1 when it is a state-owned enterprise and 0 otherwise), debt-to-asset ratio (*DAR*, defined by the proportion of total debt to total capital), and degree of competition (*DC*, defined by the Herfindahl–Hirschman index).

### 3.2. Data Sources

Based on the quasi-natural experiment involving the implementation of China’s cleaner production industry standards in 2003, this study tries to discuss the consequences of implementing this policy on the green transformation of China’s manufacturing enterprises. To start with, in terms of sample selection, considering the enterprise exit time length (mainly caused by policy expectations, which are caused by the follow-up implementation of environmental technology standards) and subsequent events (i.e., financial crisis) or policy interferences (i.e., the eleventh five year plan to eliminate backward production capacity), this study only selects Chinese manufacturing enterprises from 2000 to 2006, and on this basis, by referring Brandt et al. [44], samples in this study are further reduced by establishing and implementing the following standards: a lack of key indicators; less than 8 employees; net value of current assets or fixed assets greater than the total assets; accumulated depreciation less than the current depreciation; debt-to-asset ratio of less than 0; have implemented other similar functional policies, wages, and value-added taxes; financial fees are negative, which involves three considerations, including ensuring the integrity and availability of data required for index measurements, selecting large-scale manufacturing enterprises in normal operation, and eliminating other similar policy interference. Finally, 177,505 total observations are obtained. Moreover, according to the construction characteristics of the difference-in-difference (*DID*) model, the sample of enterprises in the three quartile industries of petroleum refining industry (2511), coking industry (2520), and leather industry (1910) are used as the treatment group, while other enterprises belong to the control group. Additionally, the data involved in this paper are all from the China Industrial Enterprise Database and China Industrial Pollution Source Key Enterprise Database. The China Industrial Pollution Source Key Enterprise Database, as the most comprehensive and reliable micro-environment database in China at present, records in detail the consumption of energy resources (such as coal, oil, natural gas, and water consumption), the discharge of waste gases (such as sulfur dioxide and chemical oxygen demand), and the number of waste resource treatment facilities in polluting enterprises. Ultimately, in this study, the required samples and data are obtained through multiple rounds of matching using the above two databases. The descriptive statistical analyses of all indicators, based on the total sample, are reflected in Table 1.

### 3.3. Empirical Model Construction

To examine the role and mechanism of environmental technology standards in the green transformation of manufacturing enterprises, on the premise of taking the implementation of cleaner production industry standards in 2003 as a quasi-natural experiment, this thesis presents the following empirical model using the difference-in-difference (*DID*) method. However, there are new changes, since the quartile industry fixed effect and year fixed effect are introduced into the original *DID* model, which can reflect the influence of the variables *treat* and *time*, respectively, which are not shown in the basic empirical model presented in this thesis. Finally, the benchmark difference-in-difference (*DID*) model in this paper is defined as:(1)$$\;\;\;\;\;\;\;\;\;\;\;W_{it}=\alpha_{0}+\beta_{1}did_{it}+\theta X_{it}+\mu_{i\;}+\gamma_{ind4}+\lambda_{jt}+\varepsilon_{it\;}\;\;\;\;\;\;\;\;\;\;\;\;\;\;\;\;\;$$
where $W_{it}$ denotes variables related to the effects and mechanism of the green transformation of manufacturing enterprises. This also includes the basic information for firm *i*’s chemical oxygen demand per unit output value (*COD*), industrial sulfur dioxide emissions (${\mathrm{SO}}_{2}$), total factor productivity indicators (*TFP_OP* and *TFP_LP*), equipment scale and capacity for treating waste gas (*GEQ* and *GCA*), sulfur dioxide (SEQ and SCA) and industrial waste water (*WEQ* and *WCA*) consumption, fixed asset investment (*INV*), depreciation (*DEP* and *DEP_R*), capital efficiency (*EFF*), clean energy utilization rate (*CLE*), and recycling water consumption rate (*REC*) in year *t*. Here, $did_{it}$ denotes the implementation of environmental technology standards by firm *i* in year *t*, with a value of one assigned when the environmental technology standards have been implemented and zero otherwise; $X_{it}$ refers to a vector of control variables that primarily relate to the enterprise characteristics; $\mu_{i}$ denotes the fixed effect of the enterprise, and its purpose is to prevent the characteristics of enterprises from changing over time; $\gamma_{ind4}$ represents the quartile industry fixed effect, and its purpose is to prevent the type of industry from changing during the observation period; $\lambda_{jt}$ refers to the fixed effect of the “dichotomous industry–year” relationship to control the exogenous shocks at the dichotomous industry level over time; $\varepsilon_{it}$ refers to the residual; $\alpha_{0}$ denotes the coefficients of the constant term; $\beta_{1}$,θ respectively represent the coefficients to be measured for the corresponding terms.

Additionally, in order to further observe the dynamic effects of environmental technology standards over time and to verify the parallel trend hypothesis for the *DID* model, this thesis, referring to Jacobson et al. [45], presents the following dynamic test model:(2)$$M_{it}=\alpha_{0}+\sum_{k\geq -3,k\neq -1}^{k=3}\delta_{k}D_{it}^{k}+\theta X_{it}+\mu_{i}+\gamma_{ind4}+\lambda_{jt}+\varepsilon_{it}$$
where $M_{it}$ denotes variables related to the effects of the green transformation of manufacturing enterprises, namely the basic information of firm *i*’s chemical oxygen demand per unit output value (*COD*) and industrial sulfur dioxide emissions (${\mathrm{SO}}_{2}$) and the total factor productivity (*TFP_OP* and *TFP_LP*) in year *t*; $D_{it}^{k}$ is a dummy variable, which indicates whether firm *i* is in the year *k* before or after the implementation of the environmental technology standards in its industry in year *t*. Specifically, $D_{it}^{k}$ equals one if $t-f_{i}=k$, otherwise it equals zero, while $f_{i}$ represents the stage of environmental technology standard implementation in the industry and *K* ranges from −3 to 3. Except for $\delta_{k}$, which represents the coefficient of $D_{it}^{k}$, the definitions of other parameters in Equation (2) show no differences with those in Equation (1).

## 4. Empirical Results and Analysis

### 4.1. Parallel Trend Test

Obviously, when trying to use the *DID* method to observe the influence and path of environmental technology standards on the green transformation of manufacturing enterprises, there is a hidden assumption that the green transformation of unregulated enterprises provides effective counterfactual changes to the green transformation of regulated enterprises. However, there have also been doubts about the above implied assumption; that is, the differences between enterprises may be directly caused by internal forces, such as pre-existing time trends, rather than external forces, such as environmental regulations. As such, here we carry out the parallel trend test before introducing the *DID* model, the results of which are presented in Figure 2.

As shown in Figure 2, we use four diagrams a, b, c, and d to display the coefficient changes of variables *COD*, ${\mathrm{SO}}_{2}$
*TFP_OP*, and *TFP_LP*, respectively, during the observation period, using Formula (2) in turn. This basically confirms the parallel trend test, meaning we are undoubtedly allowed to use the *DID* method. However, compared with the subversive impacts of environmental technology standards on variables *TFP_OP* and *TFP_LP*, which represent the economic performance, variables *COD* and ${\mathrm{SO}}_{2}$, which represent the environmental performance, are more likely to only aggravate the original effect, especially variable *COD*. The reason for this result may be due to the spillover effect of the environmental technology standards leading to the release of corresponding signals across the whole society before the formal policy proposal.

Moreover, the dynamic and positive role of the environmental technology standards is confirmed to some extent by the fact that pollution emission intensity and total factor productivity are reinforced to decrease and increase over time, respectively, since 2003.

### 4.2. Estimation Results of the Effect Test

Based on Equation (1), we can obtain the empirical results of the relationship between environmental technology standards and the green transformation of Chinese manufacturing enterprises. As shown in Table 2, columns (1) and (2) display the relationship between environmental technology standards and firm pollution emissions from the perspective of industrial oxygen emissions and sulfur dioxide emissions, respectively, while columns (3) and (4) describes the relationship between environmental technology standards and total factor productivity values of enterprises using two measurements of the total factor productivity. Furthermore, the corresponding empirical results shows that the coefficients of variables *COD* and ${\mathrm{SO}}_{2}$ are negative and significant at the 1% statistical level, while the coefficients of variables *TFP_OP* and *TFP_LP* are positive and significant at the 1% statistical level, which strongly proves that the environmental technology standards are beneficial to the green transformation of enterprises, showing that the implementation of environmental technology standards can prevent the emission of pollutants and improve the total factor productivity of enterprises.

### 4.3. Estimation Results of the Mechanism Test

Except for the exploration of the direct effects of environmental technology standards, this paper also tests the mechanism behind the environmental technology standards from the perspective of technological modifications based on Equation (1). Finally, under the consideration of the whole production process of the enterprise, this paper discusses the technological modification mechanism of the environmental technology standards from three aspects: terminal governance, capital renewal, and energy structure adjustment.

#### 4.3.1. Terminal Governance

By selecting industrial waste gas, sulfur dioxide, and industrial waste water as three kinds of common pollution emissions from manufacturing enterprises, this study investigates the impacts of environmental technology standards on the terminal governance of manufacturing enterprises from two aspects: the amount of equipment involved in pollution control and the capacity of the equipment involved in pollution control.

As shown in Table 3, columns (1)–(3) mainly describe the impacts of environmental technology standards on the amount of pollution treatment equipment, while columns (4)–(6) mainly concentrate on discussing the effects of environmental technology standards on the capacity of the pollution treatment equipment. Regarding for results, the coefficients of variables in columns (1)–(3) are all significant at the 1% statistical level, with the coefficients of variables *GEQ* and *SEQ* being positive and variable *WEQ* being negative. As such, it can be concluded that the implementation of environmental technology standards not only greatly increases the investment in waste gas and sulfur dioxide treatment equipment, but also obviously reduces the investment in industrial waste water equipment, which to some extent illustrates that the overall impacts of environmental technology standards on the volume of pollution treatment equipment is uncertain. Moreover, the coefficients of variables in columns (4)–(6) are all significantly positive. Therefore, we can say that the implementation of environmental technology standards ultimately forces enterprises to improve the capacity of their pollution treatment equipment, i.e., the environmental technology standards have a definite positive impact on the capacity of the pollution treatment equipment.

#### 4.3.2. Capital Renewal

In agreement with previous studies, which show that enterprises may avoid punishment by introducing more efficient or cleaner production technologies and other capital renewal behaviors when facing stringent environmental regulations [10,13,14,38,39], this study attempts to empirically verify this view from the perspectives of fixed asset investments, depreciation, and capital efficiency.

As shown in Table 4, in a series of regression results related to the effects of environmental technology standards on enterprise capital renewal, columns (1) and (4) respectively describe the impacts on fixed asset investment and capital efficiency, while columns (2) and (3), by using different index, show the impacts on depreciation. Finally, since the coefficients of all variables are positive and significant at the 1% statistical level, we can conclude that the implementation of environmental technology standards positively affects the capital renewal by effectively increasing the fixed asset investment, accelerating depreciation, and improving the capital efficiency.

#### 4.3.3. Resource Structure Adjustment

Existing studies have shown that compared with post-treatment, some enterprises are more willing to try to increase the environmental friendliness of their production process from the roots [11,38], and the most common way is to adjust the resource structure of an enterprise. Accordingly, based on the recognition that resource structure adjustment is an important method of technological modification, this paper constructs two indicators, *CLE* and *REC*, to confirm this view.

As shown in Table 5, the effects of environmental technology standards on the use of clean and recycled resources is reported in columns (1) and (2) in turn. As we can see, the coefficients of variables *CLE* and *REC* are positive and significant at the 1% statistical level. These empirical results prove that the implementation of environmental technology standards is conducive to the resource structure adjustment of enterprises.

## 5. Conclusions, Policy Implications, and Research Prospects

### 5.1. Conclusions

In this study, through the application of the *DID* model, under the unified analysis framework and using data from 2000 to 2006, we comprehensively investigated the role and mechanism of environmental technology standards in green transformation at the enterprise level, and accordingly obtained the following main conclusions.

Firstly, from the static perspective, environmental technology standards, as powerful exogenous forces and typical command-and-control environmental regulations, have significantly and positively affected the green transformation of Chinese manufacturing enterprises; that is, the implementation of environmental technology standards is conducive to achieving win–win goals, whereby economic performance and environmental performance can be improved simultaneously.

Secondly, from the dynamic perspective, the green transformation of enterprises in China during the period of investigation experienced sharp fluctuations. In detail, after the implementation of cleaner production industry standards in 2003, the pollution emissions intensity and total factor productivity gradually decreased and increased with time, respectively, which indicated that environmental technology standards have long-term positive impacts on the green transformation of enterprises.

Thirdly, regarding the mechanism of the environmental technology standards, this paper attempted to get overcome the inertia related to technological innovation as outlined by Porter’s hypothesis by creatively constructing a comprehensive analysis framework of the technological modification mechanism by introducing indicators of terminal governance, capital renewal, and resource structure adjustment into the research. Finally, all test results confirmed the technological modification mechanism, which was also the greatest innovation of this study.

### 5.2. Policy Implications

In addition to the theoretical contributions, based on the series of empirical results, this article also has several policy implications, as follows.

Firstly, while vigorously advocating for the advantages of market-based environmental regulations, under the background of market transformation, the government should re-examine and make rational use of command-and-control environmental regulations, especially environmental technology standards, to achieve green and sustainable development on the basis of recognizing that command-and-control environmental regulations are still dominant in China.

Secondly, the “win–win” results achieved using environmental technology standards should effectively remind the government that it needs to assess to the performance of environmental regulation tools rather than the performance of environmental regulations when designing relevant policies.

Thirdly, the obviously existence of the technological modification mechanism should force the government to objectively look at the non-inevitable relationship between environmental regulation and technological innovation, which should guide enterprises to embrace technological innovation by systematically identifying the reasons why enterprises choose technological modifications rather than technological innovation, on the premise that technological innovation is more in line with the needs of China’s economic development in the future, even if technological modifications can about win–win results in the form of pollution emission reductions and economic efficiency increases.

Finally, inspired by the dynamic effects of the environmental technology standards, enterprises should update their production strategies regularly to satisfy the changes in environmental regulations, and the government needs to consider the timeliness of environmental regulations when formulating relevant strategies. Otherwise, it will directly or indirectly reduce the incentive effect of environmental regulations in the green transformation of enterprises through the locking effect.

### 5.3. Research Prospects

Although this article has great theoretical and practical value, there still exist several limitations that can be addressed in future studies. First, considering the constraints of the data availability, only a small number of variables and indicators were employed in this study. Further expansion of the indicator system should be considered to increase the robustness of the results. Second, this article only selected manufacturing enterprises as the research sample, based on the unique contribution of manufacturing enterprises to green development, while the other industries should also be introduced into further studies, which will increase the universality of the results. Third, although this study creatively proposed and verified the technological modification mechanism of environmental regulation, it ignored the potential relationship between this and the technological innovation mechanism, which should be addressed in future research.

## Figures and Tables

**Figure 1 ijerph-19-05883-f001:**
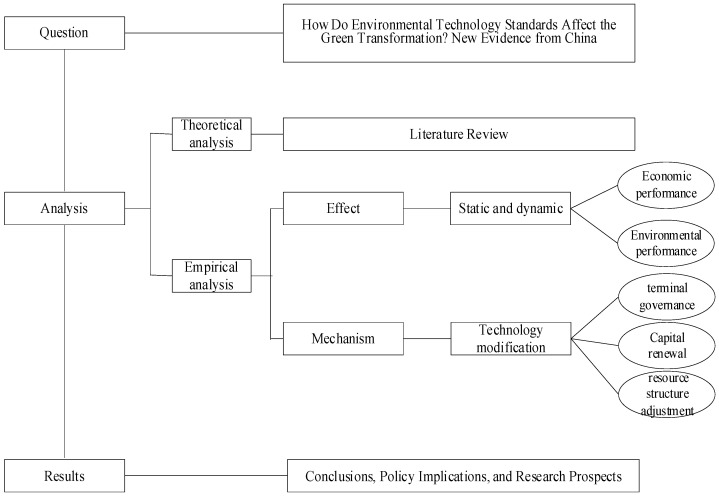
Flowchart of research steps.

**Figure 2 ijerph-19-05883-f002:**
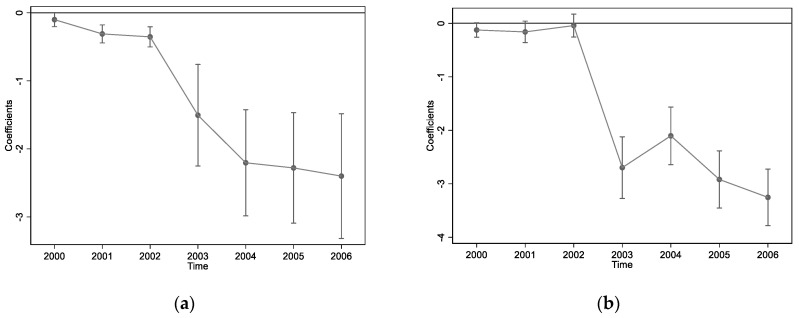
Parallel trend test: (**a**) coefficient trend of *COD*; (**b**) coefficient trend of ${\mathrm{SO}}_{2}$; (**c**) coefficient trend of *TFP_OP*; (**d**) coefficient trend of *TFP_LP*.

**Table 1 ijerph-19-05883-t001:** Statistical description of relevant variables.

Variables	Observations	Mean	S.D.	Min	Max
*DID*	177,505	0.017	0.129	0	1
*COD*	177,505	1.233	4.281	0	28.133
${\mathrm{SO}}_{2}$	177,505	2.448	5.929	0	40.259
*TFP_OP*	177,505	1.593	1.058	−7.700	7.779
*TFP_LP*	177,505	6.399	1.207	−2.859	12.617
*GEQ*	177,505	0.847	0.789	0	7.171
*SEQ*	177,505	0.166	0.413	0	4.454
*GCA*	177,505	6.234	4.449	0	21.172
*SCA*	177,505	0.515	1.375	0	14.815
*WEQ*	177,505	0.522	0.494	0	7.601
*WCA*	177,505	3.218	3.215	0	15.113
*INV*	177,505	7.635	1.931	−4.224	16.812
*DEP*	177,505	6.819	1.805	−0.121	15.144
*DEP_R*	177,505	5.999	5.231	0	33.246
*EFF*	177,505	−7.152	1.579	−14.756	2.478
*CLE*	177,505	2.113	13.669	0	100
*REC*	177,505	27.721	31.384	0	100
*EO*	177,505	0.166	0.372	0	1
*SZ*	177,505	9.360	1.679	−0.121	16.957
*ES*	177,505	0.201	0.400	0	1
*DAR*	177,505	0.646	0.292	0.025	1.577
*DC*	177,505	3.088	3.445	0.140	38.382

**Table 2 ijerph-19-05883-t002:** Estimation results of the effect test.

Variables	*COD*	$SO_{2}$	*TFP_OP*	*TFP_LP*
	(1)	(2)	(3)	(4)
*DID*	−1.705 ***	−0.340 ***	0.242 ***	0.267 ***
	(0.040)	(0.073)	(0.008)	(0.010)
*EO*	0.030	0.064	−0.042 ***	−0.013
	(0.049)	(0.056)	(0.014)	(0.014)
*SZ*	−0.115 ***	−0.250 ***	−0.235 ***	0.019
	(0.043)	(0.040)	(0.017)	(0.018)
*ES*	−0.051 *	−0.086 **	0.015 *	0.036 ***
	(0.027)	(0.039)	(0.009)	(0.009)
*DAR*	0.071	0.193 *	−0.233 ***	−0.238 ***
	(0.062)	(0.108)	(0.020)	(0.021)
*DC*	0.002	0.004	−0.002	−0.003 **
	(0.005)	(0.009)	(0.001)	(0.002)
Fixed variables	Yes	Yes	Yes	Yes
Observations	150,195	151,118	153,237	153,237
R-squared	0.712	0.612	0.664	0.747

Note: *t* statistics in parentheses; *** *p* < 0.01, ** *p* < 0.05, * *p* < 0.1.

**Table 3 ijerph-19-05883-t003:** Regression results of the terminal governance test.

Variables	*GEQ*	*SEQ*	*WEQ*	*GCA*	*SCA*	*WCA*
	(1)	(2)	(3)	(4)	(5)	(6)
*DID*	0.097 ***	0.016 ***	−0.033 ***	1.034 ***	0.029 **	0.729 ***
	(0.006)	(0.005)	(0.007)	(0.020)	(0.014)	(0.057)
*EO*	0.009	0.005	−0.001	0.010	0.006	−0.029
	(0.007)	(0.006)	(0.005)	(0.050)	(0.022)	(0.030)
*SZ*	0.034 ***	0.011 ***	0.015 ***	0.112 ***	0.021 ***	0.096 ***
	(0.006)	(0.002)	(0.002)	(0.021)	(0.007)	(0.012)
*ES*	0.007	0.001	0.005	0.022	−0.013	0.062 ***
	(0.005)	(0.004)	(0.003)	(0.041)	(0.013)	(0.016)
*DAR*	−0.004	0.002	−0.007	0.033	0.013	−0.058
	(0.013)	(0.006)	(0.007)	(0.092)	(0.026)	(0.040)
*DC*	−0.001	−0.000	−0.001	−0.008	0.000	−0.014 **
	(0.001)	(0.001)	(0.001)	(0.006)	(0.002)	(0.005)
Observations	148,071	118,090	147,416	128,160	117,840	147,492
R-squared	0.783	0.684	0.769	0.722	0.605	0.800

Note: *t* statistics in parentheses; *** *p* < 0.01, ** *p* < 0.05.

**Table 4 ijerph-19-05883-t004:** Regression results of the capital renewal test.

Variables	*INV*	*DEP*	*DEP_R*	*EFF*
	(1)	(2)	(3)	(4)
*DID*	0.282 ***	0.283 ***	1.853 ***	0.029 ***
	(0.016)	(0.011)	(0.039)	(0.001)
*EO*	−0.069 **	−0.045 ***	−0.307 ***	0.001
	(0.029)	(0.015)	(0.083)	(0.002)
*SZ*	0.854 ***	0.550 ***	−0.669 ***	−0.980 ***
	(0.018)	(0.011)	(0.059)	(0.002)
*ES*	0.027 **	0.066 ***	0.155 ***	0.005 ***
	(0.013)	(0.011)	(0.045)	(0.001)
*DAR*	−0.216 ***	0.030	0.079	−0.029 ***
	(0.039)	(0.021)	(0.104)	(0.003)
*DC*	−0.002	−0.004 **	−0.010	−0.001 ***
	(0.003)	(0.002)	(0.010)	(0.000)
Observations	91,883	145,870	156,576	153,352
R-squared	0.704	0.821	0.268	0.997

Note: *t* statistics in parentheses; *** *p* < 0.01, ** *p* < 0.05.

**Table 5 ijerph-19-05883-t005:** Regression results of the resource structure adjustment test.

Variables	*CLE*	*REC*
	(1)	(2)
*DID*	0.570 ***	2.547 ***
	(0.089)	(0.239)
*EO*	0.128	0.593
	(0.158)	(0.434)
*SZ*	0.057	0.379 *
	(0.048)	(0.226)
*ES*	−0.045	0.533 ***
	(0.111)	(0.194)
*DAR*	0.027	0.239
	(0.156)	(0.373)
*DC*	−0.015	0.106 ***
	(0.029)	(0.033)
Observations	91,634	141,008
R-squared	0.781	0.728

Note: *t* statistics in parentheses; *** *p* < 0.01, * *p* < 0.1.

## Data Availability

The data used to support the findings of this study are available from the corresponding author upon request.

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
