# Peer review of "How Do Environmental Technology Standards Affect the Green Transformation? New Evidence from China"

_ijerph, 2022, doi:10.3390/ijerph19105883_

Round 1
Reviewer 1 Report
The paper entitled “How does Environmental Technology Standard Affect the Green Transformation of Manufacturing Industry? New Evidence from China” by Xiufeng Zhang, Yan Li, Yanchao Feng describes the role and mechanism of environmental technology standard on green transformation at the enterprise level in China.
The paper is well written and interesting to read, however I see the following major issues that should be resolved before publishing this paper:
- this paper attempts to get rid of the inertia of technology innovation advocated by Porter hypothesis, and creatively constructs a comprehensive analysis framework of technology modification mechanism by introducing indicators terminal governance, capital renewal, and resource structure adjustment into research.
- In this sense, it is necessary to justify why the authors opt for these three variables (terminal governance, capital renewal, and resource structure adjustment) and not others.
- Have the authors considered differentiating between radical and incremental technological innovation?
- However, different from the exist studies, this paper innovatively adopts the way of measuring economic performance and environmental performance respectively to show the effect of green transformation, which can greatly help people to clearly identify the pure green contribution of environmental technology standard, rather than trying to build a comprehensive indicator
- How easy or difficult is it to analyze the isolated results of these indicators (identify pure green contribution)?
- To start with, in terms of sample selection, considering the length of enterprise exit time and subsequent events or policy interference, this study only selected Chinese manufacturing enterprises from 2000 to 2006,
- Taking into account that the standard started in 2003, isn't it more appropriate to study current data?
- and on this basis, samples of this study were further reduced by establishing and implementing the following standards: the lack of key indicators, the number of employees is less than 8, the net value of current assets or fixed assets is greater than the total assets, the accumulated depreciation is less than the current depreciation, the debt to asset ratio is less than 0, has implemented other similar functional policies, wages and value-added tax, financial fee are negative.
- Why these variables and not others? No justification
- Table 1 shows the variables to be analyzed. It would be appropriate to include them in an APPENDIX together with the justification of previous works that have analyzed them.
CONCLUSIONS: it would be necessary to delve deeper into the implications for policy but also from the point of view of companies
TITLE: The title is rather long. I suggest: How does Environmental Technology Standard Affect the Green Transformation?
Authors can include China and Manufacturing Industry as a key words
Author Response
1.In this sense, it is necessary to justify why the authors opt for these three variables (terminal governance, capital renewal, and resource structure adjustment) and not others.
Reply: Many thanks for your constructive suggestion. After careful consideration, we have rewritten the related contents and supplemented more details about why we choose the above variables in Dependent Variables section. The revision is given below.
“ Technology modification, as a broad concept, needs to be further decomposed in this paper, and in terms of the selection of concrete mechanisms, we mainly based on the systematic characteristics of green transformation and constructed a mechanism system of “ input-production-output”, which involves the whole production process. On this basis, in line with the existing studies, we further respectively employed energy structure, capital renewal, and terminal governance as the proxy variables corresponding to the above mechanisms. ”
2.Have the authors considered differentiating between radical and incremental technological innovation?
Reply: Many thanks for your question. And after discussion, we are likely to understand your question as “ Do you distinguish the possible relationship between incremental technological innovation and technological modification on the premise of dividing technology innovation into two categories radical and incremental technological innovation?” Our answer is “Yes”, and to avoid readers having the same questions on this issue, we have added a description of the concept of technology modification in the Introduction section. The corresponding description is presented as follows:
“On the contrary, when considering various constraints, technology modification, which is completely different from technology innovation and mainly contains the technological results that enterprise does not directly participate in the R&D process, is the best and first choice of enterprises [13,14,15]. ”
3.How easy or difficult is it to analyze the isolated results of these indicators (identify pure green contribution)?
Reply: Many thanks for your question. Generally speaking, although majority of scholars have made many breakthroughs and achievements in measuring the green transformation effect with comprehensive indicators, some scholars, like us, mainly shows the effect of green transformation more clearly and deeply from the perspectives of “emission reduction” and “ efficiency enhancement”, that is, build two kinds of dependent variables. On this basis, they also regard the effect of “ emission reduction” as pure green transformation effect in a narrow sense. Additionally, because of the isolate measurement always do not involves more complex contents, it is easy to identify the pure green contribution as a whole.
4.Taking into account that the standard started in 2003, isn't it more appropriate to study current data?
Reply: Many thanks for your question. We mainly give our explanations from the following three aspects: (1) Based on the timeliness character of a policy effect, we did not extend the observation period indefinitely. (2) Considering that cleaner production standards are implemented in stages, and the policy expectations brought by subsequent policy implementation often lead to self selection behaviors (i.e. enterprise exit), we only take the first implementation of environmental technology standards as a quasi natural experiment. Accordingly, the observation period is shortened to the next policy implementation. (3) The shorter observation period is used to avoid the interference of other events or policies, such as, financial crisis and the eleventh five year plan to eliminate backward production capacity, on the results.
Although we have explained the above question to a certain extent in the Data Sources part, here we still updated the relevant statements in more detail. The final expression is as follows.
“To start with, in terms of sample selection, considering the length of enterprise exit time ( mainly caused by policy expectations, which further born from the follow-up implementation of environmental technology standards) and subsequent events (i.e. financial crisis) or policy interference( i.e. the eleventh five year plan to eliminate backward production capacity), this study only selected Chinese manufacturing enterprises from 2000 to 2006, ”
5.Why these variables and not others? No justification
Reply: Many thanks for your constructive suggestion. To solve this problem, we added some explanatory descriptions in Data Sources section, which shown as follows.
“and on this basis, by referring Brandt et al.[44], samples of this study were further reduced by establishing and implementing the following standards: the lack of key indicators, the number of employees is less than 8, the net value of current assets or fixed assets is greater than the total assets, the accumulated depreciation is less than the current depreciation, the debt to asset ratio is less than 0, has implemented other similar functional policies, wages and value-added tax, financial fee are negative, which totally involves three kinds of considerations, including ensure the integrity and availability of data required for index measurement, select large-scale manufacturing enterprises in normal operation, and eliminate other similar policy interference.”
According to the above changes, we added the research of Brandt et al. (2012) in References.
6.Table 1 shows the variables to be analyzed. It would be appropriate to include them in an APPENDIX together with the justification of previous works that have analyzed them.
Reply: Many thanks for your high-quality suggestion. Indeed, it is an appropriate arrangement to include Table 1 in the APPENDIX. However, considering no other contents introduced in the APPENDIX, we keep Table 1 in the text. Even so, we also appreciate your considerable and constructive suggestion.
7.CONCLUSIONS: it would be necessary to delve deeper into the implications for policy but also from the point of view of companies.
Reply: Many thanks for your constructive suggestion. In fact, as the deduction and sublimation of the results of empirical analysis, policy implications have systematically taken the performance and future actions of enterprises into account. However, since we take government as the main service object of this part, the former statement, to some extent, deviates from your expectations. Thus, we followed your advice and rewrote the conclusions, with a deeper analysis from the perspective of companies.
8.TITLE: The title is rather long. I suggest: How does Environmental Technology Standard Affect the Green Transformation?
Authors can include China and Manufacturing Industry as a key words
Reply: Many thanks for your constructive suggestion. After seriously comparison, we sincerely believe that we should make some changes in our title, even if not exactly meet your expectation, and the final expression of the title is “ How does Environmental Technology Standard Affect the Green Transformation? New Evidence from China”. Additionally, we also added “ manufacturing industry” to the keywords.
Reviewer 2 Report
Dear Authors,
I have had a chance to revise your paper 'How does Environmental Technology Standard Affect the Green Transformation of Manufacturing Industry? New Evidence from China' submitted to IJERPH.
The paper addresses a potential research gap in literature and can be of potential interest to the readers of the journal. The methodology is suitable to the study objectives and the study seems sound overall.
A few remarks to improve the paper:
- In Abstract: what do you mean by ‘high-quality development of China’?
- In Introduction, line 116: change ‘thesis’ to ‘paper’
- Revise the text once again
I wish you all the best.
Kind regards,
Author Response
- In Abstract: what do you mean by‘high-quality development of China’?
Reply: Many thanks for your question. Briefly speaking, high-quality development is a new economic development model that absolutely different from the traditional high-speed development model and more in line with the current development needs. It is an active strategic choice made by China based on the domestic and foreign economic environment. In contrast, high-quality development puts more emphasis on innovation driven rather than factor driven, pays more attention to efficiency rather than speed, concerning more about the upgrading and optimization of industrial structure rather than being satisfied with staying in low-end industries chain. In addition, as a national strategy, high-quality development strategy derived from the same origin with other macro strategies, such as supply side structural reform and green transformation.
2.In Introduction, line 116: change ‘thesis’ to ‘paper’
Reply: Many thanks for your constructive suggestion. We have made corresponding modification in the article.
3.Revise the text once again
Reply: Many thanks for your careful review and constructive suggestions. Combined with the comments of other reviewers, we have revised the article according to your expectations.
Reviewer 3 Report
Dear Authors,
Thank you for the opportunity to read and review your manuscript submitted to the International Journal of Environmental Research and Public Health. After reading the manuscript, I congratulate you on the comprehensive research on environmental technology standard and its effect on green transformation. It was a pleasure to read your manuscript. Even though you did a great job, I have a recommendation for strengthening the manuscript before publication: please carefully check the language.
The research on environmental technology standard impact upon green transformation of manufacturing industry in China addressed the following questions:
- Does environmental regulation affects green transformation of manufacturing enterprises?
- How environmental regulation affects green transformation of manufacturing enterprises?
In the current environment of manufacturing companies, the impact of environmental technology standards on green transformation is a relevant object of the research. Among other vital factors in the external and internal environment that determine green transformation, environmental technology standard is the one that is rarely mentioned in the scholarship. However, this factor may be important when going towards green solutions. The relevance of the research results unfolds in the contributions to the government (re-examining the command-and-control environmental regulation, paying attention to environmental regulation tools, guiding the mechanism of technology modification to technology innovation, and updating environmental technology standard regularly).
The authors have paid sufficient attention to the research methodology that thoroughly explains the selection of the variables, data sources, and empirical model. The results of the research are explicit, main results are properly provided in tables.
The conclusions are consistent with the main results and arguments presented. They address the main research questions. The references are appropriate; the reference list includes a variety of sources in terms of time span.
Even though the manuscript is structurally and methodologically sound, there are some language issues. Therefore, the advice is to check the language carefully. While reading I have found some sentences that need proofreading, for example, lines 73, 76, 86, 134. This is the reason why I suggest minor revision.
To sum up, I would like to congratulate you on the comprehensive research on environmental technology standard and its effect on green transformation. It was a pleasure to read your manuscript.
Author Response
Even though the manuscript is structurally and methodologically sound, there are some language issues. Therefore, the advice is to check the language carefully. While reading I have found some sentences that need proofreading, for example, lines 73, 76, 86, 134. This is the reason why I suggest minor revision.
Reply: Many thanks for your careful review and constructive suggestions. After careful proofreading, we have revised the article accordingly. More details, please see the marked revision.